# Outcomes in asymptomatic, severe aortic stenosis

**Anette Borger Kvaslerud**[1,2,3]*, **Kenan Santic**[1], **Amjad Iqbal Hussain**[2,3], **Andreas Auensen**[1,2,3], **Arnt Fiane**[1,4], **Helge Skulstad**[1,2], **Lars Aaberge**[2], **Lars Gullestad**[1,2,3], **Kaspar Broch**[2,3]

1 Faculty of Medicine, University of Oslo, Oslo, Norway, 2 Department of Cardiology, Oslo University Hospital, Rikshospitalet, Norway, 3 KG Jebsen Cardiac Research Center and Center for Heart Failure Research, Oslo University Hospital, Ullevål, Norway, 4 Department of Cardiothoracic Surgery, Oslo University Hospital, Rikshospitalet, Norway

* a.b.kvaslerud@medisin.uio.no

## Abstract

### Background and aim of the study

Patients with asymptomatic, severe aortic stenosis are presumed to have a benign progno-sis. In this retrospective cohort study, we examined the natural history of contemporary patients advised against aortic valve replacement due to a perceived lack of symptoms.

### Materials and methods

We reviewed the medical records of every patient given the ICD-10-code for aortic stenosis (I35.0) at Oslo University Hospital, Rikshospitalet, between Dec 1st, 2002 and Dec 31st, 2016. Patients who were evaluated by the heart team due to severe aortic stenosis were categorized by treatment strategy. We recorded baseline data, adverse events and survival for the patients characterized as asymptomatic and for 100 age and gender matched patients scheduled for aortic valve replacement.

### Results

Of 2341 patients who were evaluated for aortic valve replacement due to severe aortic ste-nosis, 114 patients received conservative treatment due to a lack of symptoms. Asymptom-atic patients had higher mortality than patients who had aortic valve replacement, log-rank p<0.001 (mean follow-up time: 4.0 (SD: 2.5) years). Survival at 1, 2 and 3 years for the asymptomatic patients was 88%, 75% and 63%, compared with 92%, 83% and 78% in the matched patients scheduled for aortic valve replacement. 28 (25%) of the asymptomatic patients had aortic valve replacement during follow-up. Age, previous history of coronary artery disease and N-terminal pro B-type natriuretic peptide (NT-proBNP) were predictors of mortality and coronary artery disease and NT-proBNP were predictors of 3-year morbidity in asymptomatic patients.

**Data Availability Statement:** The Regional Ethics Committee in Norway (REK Sør-Øst) approved the conduction of the study. A condition for approval was that privacy concerns were respected and that

data were not made publicly available. However, excerpts of de-identified data relevant to the study can be made available upon reasonable request to Professor and Director of the Department of Cardiology at Oslo University Hospital, Rikshospitalet, Thor Edvardsen (email: thor. edvardsen@medisin.uio.no).

**Funding:** The authors received no specific funding for this work.

**Competing interests:** The authors have declared that no competing interests exist.

## Conclusions

In this retrospective study, asymptomatic patients with severe aortic stenosis who were advised against surgery had significantly higher mortality than patients who had aortic valve replacement.

## Introduction

Aortic stenosis is the most common valvular heart disease, affecting some 5% of the population older than 65 years [1]. The only definitive therapy is aortic valve replacement (AVR), and intervention for severe aortic stenosis is recommended when symptoms develop. The best management strategy for asymptomatic patients with severe aortic stenosis is the subject of ongoing debate. Current European guidelines recommend AVR for selected patients with asymptomatic aortic stenosis, namely patients with left ventricular dysfunction, abnormal exercise test, indication for other cardiac surgery, very severe aortic stenosis (defined as peak aortic flow velocity > 5.5m/s), rapid progression (defined as an increase in peak aortic flow velocity ≥ 0.3 m/s per year), markedly elevated BNP levels (> threefold normal range) or severe pulmonary hypertension (defined as a systolic pulmonary artery pressure at rest > 60mmHg), all class IIa recommendations [1]. The American guidelines recommend AVR for abnormal exercise tolerance, very severe aortic stenosis (peak aortic flow velocity > 5m/s) and rapid progression (class IIa/IIb) [2]. However, for the majority of patients with asymptomatic, severe aortic stenosis, a strategy of active surveillance is applied.

In patients with asymptomatic, severe aortic stenosis it has appeared relatively safe to follow a conservative strategy. The incidence of sudden cardiac death is estimated to be around 1% per year [3–5] as opposed to the high mortality in symptomatic patients. Considering the risk of mortality with AVR, a strategy of watchful waiting has been recommended in asymptomatic patients. More recently, however, a number of observational studies have challenged this strategy, suggesting that the incidence of sudden death might be higher than previously reported, and that most of these patients experience clinical events or require valvular surgery within two years [6–10]. The results of the first randomized trial comparing conservative strategy and surgical AVR in 145 asymptomatic patients, were in favor of early AVR. However, this study comprised a highly selected patient population with very severe aortic stenosis, young age and high prevalence of bicuspid valve [11]. Given today's low periprocedural mortality rates and particularly the advent of transcatheter AVR, early intervention has been increasingly advocated [12–14].

With this retrospective cohort study, we aimed to determine characteristics and outcomes in patients who were advised against surgery due to a perceived lack of symptoms at our tertiary centre between the years of 2002 and 2016. We wanted to examine at what attrition rate these patients come to acquire valve replacement, and if the prognosis is as good as previously reported.

## Materials and methods

### Study design and patient population

In this retrospective cohort study, we screened all patients who had been given the ICD-10 code for aortic stenosis (I35.0) at the cardiology ward at Oslo University Hospital, Rikshospitalet, between Dec 1st, 2002 and Dec 31st, 2016. By reviewing the patients' medical records we

identified every patient who had been electively admitted to our tertiary center for evaluation for AVR due to severe aortic stenosis. Patients who were evaluated by the heart team for severe aortic stenosis, were further categorized by treatment strategy: Patients who were scheduled for AVR, patients who declined surgery in spite of a recommendation to have AVR, patients who were declined from AVR due to high surgical risk and patients who were advised against AVR due to a perceived lack of symptoms.

The patients who were categorized as asymptomatic were included in this study. Inclusion criteria were age > 18 years, severe AS evaluated by the heart team for AVR, and that the patients were advised against surgery due to a perceived lack of symptoms. Exclusion criteria were contraindication to aortic valve replacement, severe extra-cardiac disease with limited expected survival, and mild to moderate aortic valve stenosis. Severe aortic stenosis was defined according to prevailing guidelines as an aortic valve area $\leq 1 cm^2$, mean pressure gradient $\geq 40$ mmHg and maximal jet velocity $\geq 4$ m/s [1].

The patients who were categorized as asymptomatic were identified based on information from the electronic patient journal. In most cases, the evaluation of symptoms was based on patient history. Only 13 patients (11%) had a cardiopulmonary exercise test.

For comparison with the asymptomatic patients, we identified 100 age and gender matched patients with severe aortic stenosis who were referred for AVR.

The endpoints for this study were all-cause mortality, subsequent AVR in patients originally advised against AVR, and the composite endpoint of major adverse cardiovascular events (MACE: all-cause mortality, stroke, transient ischemic attack (TIA), myocardial infarction and hospitalization for heart failure).

The study protocol complies with the Declaration of Helsinki and was approved by the Regional Committee for Ethics in Medicine, which waived the need for patient consent because of the retrospective nature of the study. This article follows the standards for reporting observational studies [15].

## Clinical data

Baseline clinical characteristics, biochemistry and imaging data for the asymptomatic patients and the matched group referred for AVR were obtained from the patients' medical records at Oslo University Hospital at the time of evaluation for AVR. Peripheral blood sampling was performed in the morning and the patients were non-fasting. Patients who were refused from surgery at Oslo University hospital were referred back to their local hospitals for further follow-up including routine echocardiography in accordance with prevailing guidelines. Adverse events, hospitalizations and whether the asymptomatic patients subsequently had AVR were assessed by reviewing medical records from the patients' local hospitals in the time following the initial evaluation for AVR. By March 2018, mortality data were obtained from the national Norwegian Cause of Death Registry.

## Doppler echocardiography

Transthoracic echocardiography was performed as part of the routine clinical evaluation at Oslo University Hospital, Rikshospitalet, using commercially available ultrasound scanners. The maximal aortic jet velocity was measured using continuous-wave Doppler ultrasound in multiple acoustic windows. The maximal instantaneous and mean pressure gradients across the aortic valve were measured using the time velocity integral, and the aortic valve area was calculated using the continuity equation. Left ventricular ejection fraction (LVEF) was stated in the echocardiography report. However, for a large number of patients who had normal ejection fraction, the clinician did not write the exact percentage, instead they wrote that the LVEF

was within normal values or $> 50\%$. Therefore we have reported the number of patients in each group with reduced LVEF.

## Statistical analysis

Data analyses were performed using IBM SPSS V.25 or STATA V.15. Baseline data are expressed as means with standard deviation (SD), as medians with interquartile range (IQR) or as numbers and percentages depending on distribution. Between-group differences were tested using the Student's $t$-test, Mann-Whitney U test or Pearson $\chi^2$ test when appropriate. Kaplan-Meier curves were constructed to illustrate survival among the patient categories. Differences in event-free survival rates were tested using the Cox-Mantel log-rank test. A competing-risk regression model by the *stcrreg* command in STATA was performed to evaluate associations between baseline characteristics and outcomes. The model treated AVR during follow-up as a competing risk and either death or MACE as the event of interest. Subdistribution hazard ratios (HRs) (HRs accounting for the competing risk of later AVR) were estimated with 95% Confidence Intervals (CI). Logarithmic transformation was performed to achieve normal distributions for skewed variables such as N-terminal pro-B-type natriuretic peptide (NT-proBNP). Baseline variables included in multivariable models were selected based on existing literature and specified prior to testing [1,16,17]. For mortality analyses, we included age, gender, peak aortic jet velocity, NT-proBNP, diabetes, creatinine and previous history of coronary artery disease (either necessitating percutaneous coronary intervention/coronary artery bypass surgery or medication for stable angina), and for morbidity analyses we included gender, age, NT-proBNP, diabetes and previous history of coronary artery disease. For the analysis of the best cut-off value for NT-proBNP for prediction of all-cause mortality, we used the *stroccurve package* in STATA to calculate the nearest-neighbor receiver operative characteristics (ROC) curves at 1, 2, 3 and 4 years [18]. The point on the receiver operating characteristic curve closest to (0.1) was chosen as the optimal cut-point. A *P*-value less than .05 was considered statistically significant, and all *P* values were 2-tailed.

## Results

### Clinical characteristics

Between Dec 1st, 2002 and Dec 31st, 2016, we identified 3454 patients with the ICD-10 code for aortic stenosis (Fig 1). Of these, 1006 patients did not have severe aortic stenosis, 105 patients had been evaluated or had aortic valve surgery prior to Dec 1st, 2002, and 2 patients were not Norwegian citizens and were referred to a hospital in their country of residency for further evaluation. During the period in question, the heart team at Oslo University Hospital, Rikshospitalet evaluated 2341 patients for possible AVR due to severe aortic stenosis. Of these, 1953 patients were referred for AVR, including surgical AVR as well as trans-catheter aortic valve implantation (TAVI). The remaining patients (n = 388) received conservative treatment due to either a lack of symptoms (n = 114), patient refusal in spite of a recommendation to have AVR (n = 49), or a high risk-benefit ratio or because they had comorbidities presumed to reduce life expectancy significantly (n = 225).

Baseline characteristics of the 114 patients declined from surgery due to an asymptomatic status and the 100 age and gender matched patients who had AVR are presented in Table 1. Asymptomatic patients had a mean aortic peak velocity of 4.4 (SD 0.7) m/s, mean pressure gradient of 53 (SD 16) mmHg and aortic valve area of 0.68 (SD 0.16) cm$^2$. Among the asymptomatic patients, 11 (10%) had LVEF $< 50\%$. Three patients had a LVEF of 30–35%, and 8 patients had a LVEF between 40% and 50%. Concomitant other valvular heart disease was present in 20 (18%) of the asymptomatic patients with the following distribution: moderate aortic

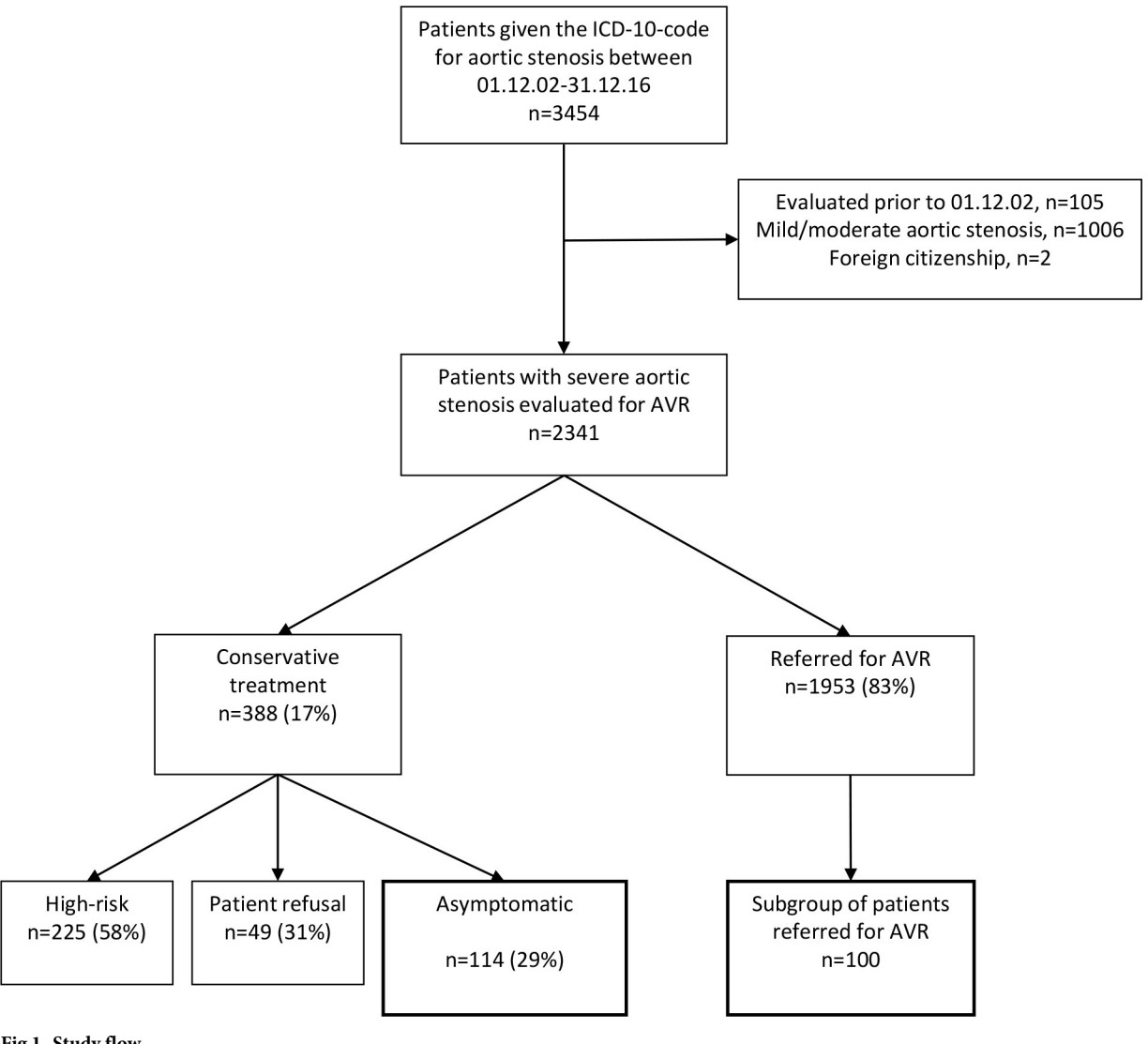

**Fig 1. Study flow.**

regurgitation: 8, moderate pulmonary regurgitation: 1, moderate mitral regurgitation: 2, moderate tricuspid regurgitation: 3, moderate to severe tricuspid regurgitation: 3, moderate mitral stenosis: 2. None of the asymptomatic patients had been diagnosed with cardiomyopathy.

## Aortic valve replacement

Of the 114 patients categorized as asymptomatic, 28 patients (25%) subsequently had AVR at a median of 1.6 (IQR 1.1–2.8) years after they were initially advised against surgery. Of these, 23 patients had surgical AVR and 5 patients had TAVI. Coronary artery bypass surgery was performed in 4 patients at the time of surgery. Two patients were scheduled for surgery but died on the waiting list. The first patient was accepted for surgery at re-evaluation 51 days after first evaluation due to syncope, but died of myocardial infarction 7 days later. The second patient was accepted for surgery 94 days after the initial evaluation due to emerging symptoms of dyspnea. He died of stroke 10 days after he was accepted for surgery. Another 18 patients developed symptoms during follow-up but were considered too comorbid and fragile and were

**Table 1.** Baseline characteristics of patients with asymptomatic severe aortic stenosis compared to 100 age and gender matched patients with severe aortic stenosis referred for aortic valve replacement.

| Variables, units | Patients with severe aortic stenosis | | |
|---|---|---|---|
| | Asymptomatic n = 114 | Referred for surgery n = 100 | p-value |
| **Demography** | | | |
| Age, years, median (IQR) | 83 (77–87) | 81 (75–85) | 0.12 |
| Male sex, n (%) | 48 (42) | 42 (42) | 0.99 |
| Married or partner, n (%) | 50 (44) | 48 (48) | 0.76 |
| Body mass index, kg/m2 | 24.8 (4.6) | 25.4 (4.3) | 0.27 |
| Current smoker, n (%) | 2 (2) | 13 (13) | **<0.001** |
| **Medical history** | | | |
| Hypertension, n (%) | 49 (43) | 48 (48) | 0.46 |
| Atrial fibrillation/flutter, all types | 27 (24) | 25 (25) | 0.82 |
| Diabetes mellitus type I and I, n (%) | 20 (18) | 10 (10) | 0.11 |
| History of coronary artery disease, n (%) | 18 (16) | 20 (20) | 0.42 |
| Pacemaker, n (%) | 6 (5) | 3 (3) | 0.41 |
| Pulmonary disease, n (%) | 12 (11) | 17 (17) | 0.17 |
| Previous history of cancer, n (%) | 14 (12) | 17 (17) | 0.33 |
| **Medication** | | | |
| Beta blocker | 48 (42) | 51 (51) | 0.19 |
| ACEi/ARB | 44 (39) | 54 (54) | **0.024** |
| Calcium antagonist | 23 (20) | 22 (22) | 0.74 |
| Cholesterol lowering agent | 51 (45) | 59 (59) | **0.037** |
| Nitrates | 4 (4) | 9 (9) | 0.093 |
| Anticoagulants | 24 (21) | 21 (21) | 0.99 |
| Platelet inhibitor | 53 (47) | 54 (54) | 0.27 |
| Diuretics | 54 (48) | 35 (35) | 0.067 |
| Diabetes medication | 15 (13) | 8 (8%) | 0.19 |
| **Clinical findings** | | | |
| Systolic blood pressure, mmHg | 147 (26) | 145 (22) | 0.67 |
| Diastolic blood pressure, mmHg | 74 (13) | 75 (12) | 0.92 |
| **Biochemistry** | | | |
| Haemoglobin, g/dL | 13.4 (1.4) | 13.4 (1.6) | 0.89 |
| Cholesterol, mmol/L | 5.0 (1.2) | 5.1 (1.1) | 0.65 |
| LDL Cholesterol, mmol/L | 2.9 (1.0) | 2.9 (0.92) | 0.98 |
| NT-pro-BNP, ng/L, median (IQR) | 1006 (410–2579) | 973 (321–1928) | 0.50 |
| Troponin T, ng/mL, median (IQR) | 15 (5–26) | 14 (10–25) | 0.096 |
| Creatinine, μmol/L | 88 (28) | 88 (30) | 0.89 |
| eGFR<60 mL/min, n (%) | 37 (32) | 38 (38) | 0.52 |
| eGFR mL/min in patients with eGFR < 60 | 48 (13) | 48 (10) | 0.97 |
| HbA1c, % | 5.9 (1.1) | 5.9 (0.6) | 0.79 |
| **Echocardiographic measures** | | | |
| Aortic peak velocity, m/s | 4.4 (0.7) | 4.9 (0.7) | **<0.001** |
| Aortic mean gradient, mm Hg | 53 (16) | 59 (19) | **0.008** |
| Aortic valve area, cm$^2$ | 0.68 (0.16) | 0.65 (0.20) | 0.27 |
| LVEF<50%, n (%) | 11 (9.6) | 16 (16) | 0.10 |
| TRPG | 33 (10) | 33 (10) | 0.96 |
| SWTd, cm | 1.2 (0.2) | 1.1 (0.2) | **<0.001** |
| LVIDd, cm | 4.9 (0.6) | 5.0 (0.7) | 0.34 |

(*Continued*)

**Table 1.** (Continued)

| | Patients with severe aortic stenosis | | |
|---|---|---|---|
| Variables, units | Asymptomatic n = 114 | Referred for surgery n = 100 | p-value |
| Concomitant moderate to severe other valvular heart disease, n (%) | 20 (18) | 24 (24) | 0.24 |

The numbers are mean (S.D), frequency (%), or medians (interquartile range). P-values for comparison of results of patients perceived to be asymptomatic and of patients referred for surgery.

Abbreviations: ACEi, ACE inhibitor; ARB, angiotensin receptor blocker; CRP, C-reactive protein; NT-proBNP, N-terminal pro-B-type natriuretic peptide; LVEF, left ventricular ejection fraction; TRPG, tricuspid regurgitation pressure gradient; SWTd, septal wall thickness at end-diastole; LVIDd, left ventricular internal diameter at end-diastole.

declined from surgery due to high surgical risk on renewed evaluation. The renewed evaluation of these 18 patients was performed on average 2.5 (SD 1.7) years after the initial evaluation, and the most common comorbidities/factors that contributed to the decision to reject the patient from surgery was age, chronic obstructive pulmonary disease, TIA/stroke, kidney disease, aortic aneurism, cognitive impairment, cancer, frailty and impaired gait function. The patients who subsequently had AVR were significantly younger than the ones who remained on conservative treatment (median age, 75 years (IQR 64–81) versus 85 years (80–88), respectively, p<0.001).

## Survival

The mean duration of follow-up in all 2341 patients was 5.4 (SD 3.5) years (median 5.1 years, IQR 2.6–7.8). During this period, 1126 patients (48%) died. 73 of the 114 patients (64%) initially characterized as asymptomatic died. Survival at 1, 2 and 5 years for the asymptomatic patients was 88%, 75% and 39% respectively, compared with 91%, 87% and 74% in patients who were referred for AVR. The log-rank p-value for comparison of the distribution of survival times between the two groups was <0.001 (Fig 2). The patients referred for surgery were significantly younger than the patients categorized as asymptomatic (median 76 years (IQR 68–82) versus 83 (IQR 77–87), p<0.001). We therefore compared survival rates between the asymptomatic patients and age—and gender matched patients who were referred for AVR from the outset for a mean duration of follow-up of 4.0 (SD 2.5) years (median 4.1 years, IQR 2.0–5.4). For the asymptomatic patients, the survival at 1, 2 and 3 years was 89%, 75% and 63% respectively, compared with 92%, 83% and 78% in the patients referred for AVR. Log-rank p was <0.001 (Fig 3).

In the 2012 version of the ESC (European Society of Cardiology) Guidelines for the management of valvular heart disease [19], a new class IIa recommendation was included stating that AVR should be considered in very severe, asymptomatic aortic stenosis defined by a peak transvalvular velocity >5.5 m/s. Of the patients evaluated prior to 2012, 6 patients had transvalvular velocities >5.5 m/s. The log-rank p-value for comparison of the survival curves between asymptomatic patients and the matched patients referred to surgery when these 6 patients were excluded was <0.001.

LVEF was reduced in 11 patients (10%) who were advised against surgery due to a lack of symptoms. When we omitted these patients from the survival analyses, survival at 1, 2 and 3 years for the asymptomatic patients was 90%, 78% and 66%, respectively. The survival remained significantly worse than for the matched patients who were referred to surgery, regardless of their LVEF (log-rank p <0.001).The patients who either were declined from surgery due to high-risk profile or who refused to undergo surgery had significantly higher mortality than the asymptomatic patients, log- rank p<0.001 and p = 0.05 respectively (Fig 2).

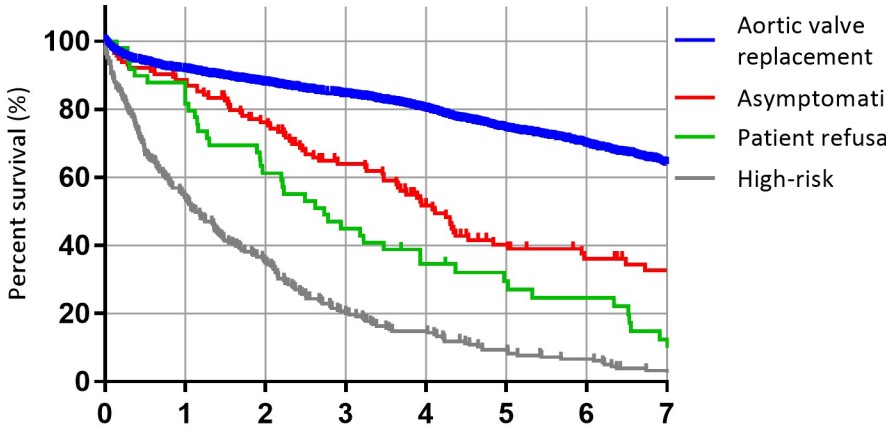

**Fig 2. Survival analysis.** Kaplan-Meier curve reflecting survival in patients with severe aortic stenosis dependent on treatment allocation.

There was no age difference between these patients and the patients perceived to be asymptomatic.

Competing risk regression analysis identified age, previous history of coronary artery disease, and NT-proBNP as predictors of mortality in patients with asymptomatic, severe aortic

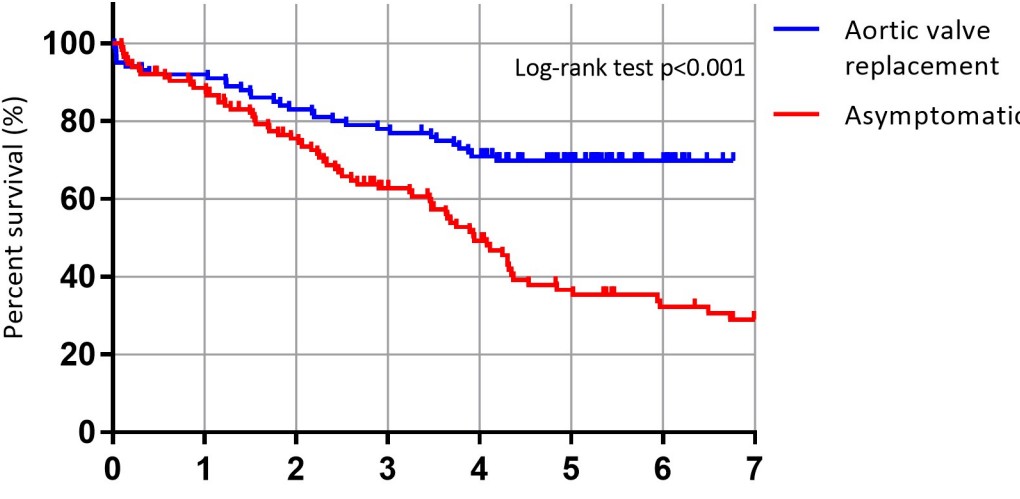

**Fig 3. Survival analysis.** Kaplan-Meier curve reflection survival in asymptomatic patients with severe aortic stenosis compared to 100 matched patients referred for aortic valve replacement.

**Table 2. Competing risk regression analysis of all-cause mortality in 114 patients with asymptomatic, severe aortic stenosis (competing event: Aortic valve replacement during follow-up).**

| Variables | SHR[a] | 95% CI | p |
|---|---|---|---|
| Male gender | 1.73 | 0.70, 4.26 | 0.23 |
| Age, per 1 year | 1.14 | 1.03, 1.27 | **0.012** |
| Diabetes | 0.98 | 0.34, 2.81 | 0.97 |
| Peak aortic velocity, per m/s | 1.47 | 0.92, 2.35 | 0.104 |
| NT- proBNP, log ng/L | 2.26 | 1.04, 4.89 | **0.039** |
| Previous history of coronary artery disease | 3.08 | 1.19, 7.96 | **0.020** |
| Creatinine, μmol/l | 1.01 | 0.99, 1.02 | 0.071 |

[a] SHR subhazard ratios, similar to hazard ratios (HR) from the classic Cox regression.

Abbreviations: NT-proBNP, N-terminal Pro-B-Type Natriuretic Peptide.

stenosis when AVR during follow-up was considered a competing event (Table 2). Calculating the ROC curves for time-dependent outcomes, we found that the best cut-off value for NT-proBNP for prediction of all-cause death after 1, 2, 3 and 4 years was 1082 ng/L (128 pmol/L) (2 years: AUC 0.66, sensitivity 0.73, specificity 0.64; 3 years: AUC 0.73, sensitivity 0.70, specificity 0.71; 4 years: AUC 0.83, sensitivity 0.70, specificity 0.83). Asymptomatic patients with NT-proBNP>1082 ng/L had higher mortality rates than patients with NT-proBNP ≤1082 ng/L, log-rank p<0.01. Likewise, patients who had AVR had higher mortality rates if their NT-proBNP was >1082 ng/L, log-rank p = 0.02.

The cause of death was recorded in the National Cause of Death Registry for 70 of the 73 asymptomatic patients who died. 31 deaths (44%) were cardiovascular and another 2 patients died suddenly from an unknown cause. Among the remaining 37 patients, the causes of death were as follows: pulmonary disease including pneumonia: 10, malignancy: 6, stroke: 6, kidney disease: 4, dementia: 4, infection other than pneumonia: 3, aortic dissection: 1, motor neuron disease: 1, diabetes mellitus: 1, bowel disease: 1. For the remaining 3 patients the death dates were too recent for the causes of death to be registered.

Thirty patients who had AVR died during follow up. Ten deaths (33%) were cardiovascular, eight died of malignancy, and six of infections. Two patients died of aortic dissection/aneurism, one of dementia, one of stroke, and two of bowel disease.

## Adverse events

A MACE occurred in 21 of the 114 asymptomatic patients (18%) in the year after the evaluation by the heart team. Among the matched patients, 16 (16%) experienced a MACE. There was no difference with regard to one-year MACE between the asymptomatic patients and the matched controls, log-rank p = 0.79.

During the first three years after the evaluation by the heart team, 56 (49%) asymptomatic patients experienced a MACE. NT-proBNP and previous history of coronary artery disease were associated with 3 year MACE on multivariable competing risk regression analysis (Table 3).

## Discussion

The main finding of our study is that patients with severe aortic stenosis who were advised against AVR due to a perceived lack of symptoms, and who did not have any other indication for surgery nor contradiction to surgery according to the prevailing guidelines, had significantly higher mortality than age and gender matched patients who were referred for AVR. A

**Table 3. Competing risk regression analysis of 3 year MACE in 114 patients with asymptomatic, severe aortic stenosis (competing event: Aortic valve replacement during follow-up).**

| Variables | SHR[a] | 95% CI | P |
|---|---|---|---|
| Male gender | 1.72 | 0.92, 3.21 | 0.087 |
| Age, per 1 year | 1.05 | 0.99, 1.11 | 0.096 |
| Diabetes | 0.73 | 0.29, 1.84 | 0.50 |
| NT- proBNP, log ng/L | 2.75 | 1.28, 5.95 | **0.010** |
| Previous history of coronary artery disease | 2.32 | 1.18, 4.59 | **0.015** |

[a] SHR subhazard ratios, similar to hazard ratios (HR) from the classic Cox regression.

Abbreviations: MACE, Major adverse cardiovascular events; NT-proBNP, N-terminal Pro-B-Type Natriuretic Peptide.

large number of the patients who developed symptoms during follow-up were not candidates for surgery due to a high-risk profile at reevaluation. Age, previous history of coronary artery disease, and NT-proBNP at the time of evaluation for AVR were independent predictors of long-term mortality on competing risk regression analysis, and NT-proBNP and previous history of coronary artery disease were independent predictors of 3-year morbidity.

The management of patients with asymptomatic, severe aortic stenosis remains challenging. Current guidelines are based on observational and retrospective studies, many of which include only a limited number of patients and were performed when the patients presenting with aortic stenosis were significantly younger and the operative risk higher than what we see today. In 1968, when Ross and Braunwald published their landmark study on the dramatic prognosis in aortic stenosis after symptom onset [5], rheumatic disease and congenital disorders were the predominant causes of aortic stenosis. The epidemiology of aortic stenosis today, however, is dominated by octogenarians with degenerative valvular disease. No pharmacological treatment can improve outcome or delay disease progression. AVR, either surgical or transcatheter, is the only treatment that improves survival. The timing of the intervention is therefore crucial. In the last decade, several observational studies and one randomized study have suggested that the natural history of asymptomatic, severe aortic stenosis may not be as benign as previously thought, challenging the management strategy of active surveillance.

As many as 37–46% of patients with severe aortic stenosis are asymptomatic [17,20]. In our study, only 114 (4.5%) of the 2454 patients who were evaluated by the heart team for AVR due to severe aortic stenosis, were advised against surgery due to a lack of symptoms. This low proportion is mainly explained by the fact that the patients included in this study had been referred to our tertiary hospital for evaluation for AVR. Furthermore, asymptomatic patients who had AVR for other reasons than symptoms of severe aortic stenosis, e.g. indication for other cardiac surgery or left ventricular dysfunction, were classified in the group of patients who were referred for AVR. Our intention with this study was to assess the prognostic implication of advising patients against surgery due to an apparent lack of symptoms among patients who have been referred to a tertiary center. In this respect, we are confident that the patient population is representative.

The verification of symptoms, and particularly the lack thereof, can be difficult. Elderly patients often lead sedentary lives, have many co-morbidities that may restrain physical activity [21], and often under-report symptoms [22]. Our results suggest that we should question the strategy of active surveillance in contemporary, asymptomatic patients with severe aortic stenosis. Particularly in the elderly, where symptoms are difficult to assess and where there is a risk that when symptoms emerge the patients are no longer candidates for surgery, we suggest

that early intervention should be considered. The current low periprocedural mortality rates, the advent of TAVI, and the occurrence of sudden death without preceding symptoms in patients with severe aortic stenosis [20], are arguments for early intervention.

Increased NT-proBNP was associated with a higher risk of adverse events. This is consistent with what have been recently shown in a study by Nakatsuma et al. [23] and supports the incorporation of BNP into the management of asymptomatic patients with severe aortic stenosis.

We are awaiting the results of another five ongoing randomized clinical trials on early AVR versus conservative treatment for asymptomatic patients with severe aortic stenosis (AVATAR (NCT02436655), EVoLVeD (NCT03094143), EASY-AS (NCT04204915), ESTIMATE (NCT02627391) and EARLY TAVR (NCT03042104)).

## Study limitations

The main limitation of this study is the selected population. We included patients referred for evaluation for AVR at our tertiary centre. However, the majority of asymptomatic patients are followed locally and referred to a tertiary centre only when there is indication for intervention, such as the development of symptoms. These patients were not included in this study. This might result in a selection bias, as the patients who are referred for evaluation at a tertiary care center most likely differ from those who are followed up locally. In addition, the number of patients who were asymptomatic was limited. Another limitation of this study is its retrospective, observational nature. The reliability of symptom assessment is always associated with uncertainty, and the choice of management strategy is at the discretion of the physician and the patient. The 18 patients who later developed symptoms but were declined from surgery at renewed evaluation, were originally rejected because they were deemed to be asymptomatic. However, there is a chance for selection bias as one could imagine that at higher age and frailty and due to comorbidity, surgeons are reluctant to operate and use the label "asymptomatic" to justify recommendation against surgery. Furthermore, only a minority of the patients had an exercise test. Finally, this was a single-centre study, which may reduce the generalizability of the results.

## Conclusions

The optimal management strategy for asymptomatic patients with severe aortic stenosis remains unclear. In our retrospective study, patients who were advised against surgery at our tertiary care hospital due to a perceived lack of symptoms had significantly higher mortality than patients referred for AVR.

## Acknowledgments

**Meeting presentation:** Preliminary results from this study were presented as a poster presentation at the ESC Heart Failure Congress in Athens, Greece, on the 26th of May 2019.

## Author Contributions

**Conceptualization:** Anette Borger Kvaslerud, Arnt Fiane, Helge Skulstad, Lars Aaberge, Lars Gullestad, Kaspar Broch.

**Data curation:** Anette Borger Kvaslerud, Kenan Santic, Amjad Iqbal Hussain, Andreas Auensen, Lars Gullestad, Kaspar Broch.

**Formal analysis:** Anette Borger Kvaslerud, Amjad Iqbal Hussain.

**Funding acquisition:** Lars Gullestad.

**Investigation:** Anette Borger Kvaslerud, Kenan Santic, Andreas Auensen.

**Methodology:** Anette Borger Kvaslerud, Kenan Santic, Amjad Iqbal Hussain, Andreas Auensen, Arnt Fiane, Helge Skulstad, Lars Aaberge, Lars Gullestad.

**Project administration:** Anette Borger Kvaslerud, Kenan Santic, Lars Gullestad, Kaspar Broch.

**Supervision:** Helge Skulstad, Lars Aaberge, Lars Gullestad, Kaspar Broch.

**Validation:** Arnt Fiane.

**Visualization:** Anette Borger Kvaslerud.

**Writing – original draft:** Anette Borger Kvaslerud, Kaspar Broch.

**Writing – review & editing:** Anette Borger Kvaslerud, Kenan Santic, Amjad Iqbal Hussain, Andreas Auensen, Arnt Fiane, Helge Skulstad, Lars Aaberge, Lars Gullestad, Kaspar Broch.

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
