## [Decision Letter · Decision Letter 0]

18 Jan 2021

PONE-D-20-37902

Patient characteristics and outcomes in asymptomatic, severe aortic stenosis

PLOS ONE

Dear Dr. Kvaslerud,

Thank you for submitting your manuscript to PLOS ONE. After careful consideration, we feel that it has merit but does not fully meet PLOS ONE’s publication criteria as it currently stands. Therefore, we invite you to submit a revised version of the manuscript that addresses the points raised during the review process.

The manuscript has been carefully evaluated by 2 external reviewers and they found the manuscript potentially of interest. However, the referees have required additional information and clarifications from the authors that need to be provided.

We look forward to receiving your revised manuscript.

Kind regards,

Claudio Passino, MD

Academic Editor

PLOS ONE

Journal Requirements:

2. We noted in your submission details that a portion of your manuscript may have been presented or published elsewhere.

"An abstract from this manuscript has been presented at the ESC Heart Failure Congress in Athens, Greece, on the 26th of May 2019"

Please clarify whether this conference proceeding was peer-reviewed and formally published. If this work was previously peer-reviewed and published, in the cover letter please provide the reason that this work does not constitute dual publication and should be included in the current manuscript.

Reviewers' comments:

Reviewer's Responses to Questions

**Comments to the Author**

1. Is the manuscript technically sound, and do the data support the conclusions?

Reviewer #1: Yes

Reviewer #2: Partly

2. Has the statistical analysis been performed appropriately and rigorously? 

Reviewer #1: Yes

Reviewer #2: Yes

3. Have the authors made all data underlying the findings in their manuscript fully available?

Reviewer #1: Yes

Reviewer #2: Yes

4. Is the manuscript presented in an intelligible fashion and written in standard English?

Reviewer #1: Yes

Reviewer #2: Yes

5. Review Comments to the Author

Reviewer #1: A very interesting an important research study. Did the authors consider performing propensity score matching between the asymptomatic severe AS patients and the matched controls. How did the authors deal with confounding factors such as baseline characteristics?

Did all asymptomatic severe AS patients have a LVEF >55%? I think patients with LVEF <55% should be excluded as asymptomatic severe AS patients were heart failure should be considered for AVR/TAVI.

Reviewer #2: Poor description of method. Paper deals with a very interesting topic, although Authors are to be commended for their attempt to limit the influence of potential confounders (review comments attached below)

6. PLOS authors have the option to publish the peer review history of their article (what does this mean?). If published, this will include your full peer review and any attached files.

Reviewer #1: **Yes: **Crochan J O'Sullivan

Reviewer #2: No

---

## [Author Response · Author response to Decision Letter 0]

24 Feb 2021

Response to reviewers

Journal Requirements:

We confirm that our manuscript conforms to the PLOS ONE’s style requirements.

2. We noted in your submission details that a portion of your manuscript may have been presented or published elsewhere.

"An abstract from this manuscript has been presented at the ESC Heart Failure Congress in Athens, Greece, on the 26th of May 2019"

Please clarify whether this conference proceeding was peer-reviewed and formally published. If this work was previously peer-reviewed and published, in the cover letter please provide the reason that this work does not constitute dual publication and should be included in the current manuscript.

The abstract was presented as a poster at the ESC Heart Failure Congress in May 2019 and contained preliminary data from this study, with no depth of knowledge dissemination. Furthermore, this poster did not present the full results as presented in this article. This abstract was not a peer-reviewed publication, and did not contain enough information to be evaluated as such. In accordance with the ICMJE, we are of the opinion that the publication of results previously presented as a poster displayed at a scientific meeting should not be considered duplicate publication (1). 

1. ICMJE. Overlapping publications: International Committee of Medical Journal Editors; 2021 [cited 2021 02.02.]. Available from: http://www.icmje.org/recommendations/browse/publishing-and-editorial-issues/overlapping-publications.html.

The data that support the findings of this study is derived from human research participant data which is potentially identifiable and sensitive. Due to legal restrictions and in compliance with regulations by the Regional Ethics Committee in Norway (REK Sør-øst) and the local privacy representative (Personvernombud) at Oslo University Hospital, the data cannot be made publicly available. Some of the data can be made available upon reasonable request. We propose to include the following statement under Data Availability where we have included contact information to the data access committee:

“Data Availability: The Regional Ethics Committee in Norway (REK Sør-Øst) approved the conduction of the study. A condition for approval was that privacy concerns were respected and that data were not made publicly available. However, excerpts of de-identified data relevant to the study can be made available upon reasonable request to Professor and Director of the Department of Cardiology at Oslo University Hospital, Rikshospitalet, Thor Edvardsen (email: thor.edvardsen@medisin.uio.no).

Response to reviewers

Reviewer's Responses to Questions

1. Is the manuscript technically sound, and do the data support the conclusions? 

Reviewer #1: Yes

Reviewer #2: Partly

2. Has the statistical analysis been performed appropriately and rigorously? 

Reviewer #1: Yes

Reviewer #2: Yes

3. Have the authors made all data underlying the findings in their manuscript fully available?

Reviewer #1: Yes

Reviewer #2: Yes

4. Is the manuscript presented in an intelligible fashion and written in standard English?

Reviewer #1: Yes

Reviewer #2: Yes

5. Review Comments to the Author

Reviewer #1: 

A very interesting and important research study. Did the authors consider performing propensity score matching between the asymptomatic severe AS patients and the matched controls. How did the authors deal with confounding factors such as baseline characteristics?

We thank the reviewer for this highly relevant comment. We agree that propensity score matching between the two groups would be a reasonable method to adjust for baseline differences. Unfortunately, due to the retrospective nature of this study, data availability was limited. All data were obtained from the electronic patient journal, and data were in many cases incomplete. We therefore do not have the data to perform propensity score matching. Instead, the groups are matched with regard to the most important confounders, age and gender. In table 1, we have presented the baseline data that was available to us. We have revised this table and included more variables allowing for better comparison between the two groups, including left ventricular ejection fraction (LVEF) and left ventricular dimensions. 

Did all asymptomatic severe AS patients have a LVEF >55%? I think patients with LVEF <55% should be excluded as asymptomatic severe AS patients were heart failure should be considered for AVR/TAVI.

For most patients, the LVEF was stated as >50 % or normal in the echocardiography report. However, 11/114 patients (10 %) had reduced LVEF. We have included this information both in table 1 and in the text:

“Among the asymptomatic patients, 11 (10 %) had LVEF < 50 %. Three patients had a LVEF of 30-35 %, and 8 patients had a LVEF between 40 and 50 %.”

We agree with the reviewer that these patients could be considered for exclusion from the analyses. According to the ESC 2017 guidelines, systolic dysfunction (LVEF <50%) “not due to another cause” qualifies for a class IC recommendation for AVR/TAVI. However, most of these patients had previous coronary artery disease and the reduced LVEF did not mean that they had indication or AVR/TAVI. 

To account for this possible bias we have included an analysis on survival where we omit the patients with reduced LVEF: 

 “LVEF was reduced in 11 patients (10 %) who were advised against surgery due to a lack of symptoms. When we omitted these patients from the survival analyses, survival at 1, 2 and 3 years for the asymptomatic patients was 90%, 78% and 66%, respectively. The survival remained significantly worse than for the matched patients who were referred to surgery, regardless of their LVEF (log-rank p <0.001). “

Reviewer #2: Poor description of method. Paper deals with a very interesting topic, although Authors are to be commended for their attempt to limit the influence of potential confounders (review comments attached below)

Attachment

Patient characteristics and outcomes in asymptomatic, severe aortic stenosis 

In the present paper Kvaslerud and Colleagues attempt to compare natural history of asymptomatic patients with severe aortic stenosis submitted to a conservative treatment with symptomatic severe aortic stenosis treated with aortic valve replacement. They reviewed medical records of every patient given the ICD-10-code for aortic stenosis (I35.0) at Oslo University Hospital between Dec 2002 and Dec 2016. Patients who were evaluated by the heart team due to severe aortic stenosis were categorized by treatment strategy; they recorded baseline data, adverse events and survival in the group of asymptomatic patients and in a comparison group of 100 age and gender matched patients scheduled for aortic valve replacement (AVR). Authors found that in 2341 patients who were evaluated for aortic valve replacement due to severe aortic stenosis, 114 patients received conservative treatment due to a lack of symptoms. Asymptomatic patients had higher mortality than patients who had AVR. Survival at 1, 2 and 3 years for the asymptomatic patients was 88%, 75% and 63%, compared with 92%, 83% and 78% in the matched patients scheduled for AVR. Age, previous history of coronary artery disease and N-terminal pro B-type natriuretic peptide (NT-proBNP) were predictors of mortality while coronary artery disease and NT-proBNP were predictors of 3-year morbidity in asymptomatic patients. 

The optimal timing of AVR in patients with severe aortic stenosis remains a commonly encountered clinical challenge in cardiovascular medicine. Practice guidelines give AVR a class I recommendation in patients with severe aortic stenosis and symptoms of heart failure, angina, or syncope. However, watchful waiting for symptoms is associated with its own risks: sudden death is estimated to affect 1%/y of asymptomatic patients with isolated severe aortic stenosis. Some patients moreover may not be recognized symptoms as related to aortic stenosis; even the straightforward presence of symptoms, especially in the elderly patient, may be difficult to ascertain. 

With these considerations, guidelines have added a few additional circumstances in which AVR is reasonable for the asymptomatic patient with severe aortic stenosis; these include left ventricular dysfunction, (defined as ejection fraction <50%), cardiac surgery for other indications, reduced exercise capacity or exertional hypotension demonstrated by exercise testing, or “very severe” aortic stenosis. With the limit of ruling out these recommendations, the paper deals with a very interesting topic, although Authors are to be commended for their attempt to limit the influence of potential confounders. 

Major points 

- Authors should make a wider description of methods; in a retrospective study evaluating the comparison between two different strategies, watchful waiting and surgical approach, characterization of patients must be more detailed, in particular in table 1 (baselines characteristics) should be inserted comorbidities (such as chronic obstructed pulmonary disease, cancer, chronic renal disease...) and frailty. Recent studies have demonstrated frailty and other geriatric syndromes can identify the most vulnerable patients, thus allowing to better estimate the individual prognosis in terms of disability, readmission and mortality. Such information should also be considered in the decision-making process, ensuring the benefit derived from each treatment in each patient, optimizing the resources and avoiding futility. In the present study a not negligible percentage, 15.7% of patient population, developed symptoms during follow-up, but were considered too comorbid and fragile and for these reasons were declined from surgery due to high surgical risk on renewed evaluation 

We thank the reviewer for this constructive and highly reasonable feedback. Prompted by the reviewers’ remarks, we have made several changes to the methods section to better describe the study population and data acquisition. In the revised Table 1, we have included data on comorbidities including pulmonary disease, cancer, eGFR and the number of patients with pacemaker. We also provide data on left ventricular dimensions, concomitant valvular heart disease, left ventricular ejection fraction and tricuspid regurgitation pressure gradient. 

We recognize that the retrospective study design has innate limitations and potential for biases due to selection, classification etc. There will always be uncertainties associated with the results from such studies, and we have therefor tried not to over-interpret our results. We have mentioned these shortcomings in the limitation section. To account for the fact that there might be unmeasured or unmeasurable factors that might confound the results, such as frailty and comorbidity, we have written the following in the limitation section: 

“there is a chance for selection bias as one could imagine that at higher age and frailty and due to comorbidity, surgeons are reluctant to operate and use the label “asymptomatic” to justify recommendation against surgery.”

At the time our patients were considered for aortic valve replacement, TAVI was in its infancy. Hence patients who were deemed too fragile for open heart surgery received conservative treatment. Today, some of these patients might have been recommended for TAVI. With the advent of TAVI, even older patients are evaluated for valve replacement and especially in these patients assessment of frailty is crucial. Therefore, at our hospital, assessment of frailty has been implemented on all patients who are presently evaluated for TAVI. Unfortunately however, this was not the case prior to 2016. A possible remark on this issue would be that the patients that were excluded from surgery due to a high risk profile could today be evaluated for TAVI, and possible their outcomes could be improved. This of course, is only speculative. 

- Crucial clinical characteristics related to cardiovascular sphere are missing: for example, coexisting cardiomyopathy, ventricular hypertrophy, valvular disease, arrhythmic burden and the presence of a pacemaker/defibrillator. Finally, also echocardiographic measures showed in table 1 are too sketchy; in this description a lot of relevant data are missing, as well as right ventricular dysfunction, diastolic function, estimated systolic pulmonary pressures. 

We agree that this information is highly relevant. In the revised Table 1, we have added variables to better characterize the two groups. These variables include LVEF, left ventricular dimensions and pacemakers. Unfortunately, we do not possess diastolic measures. We agree with the reviewer that there still are several other variables of interest, which we unfortunately lack full report of due to the retrospective character of the data. 

 - Evaluation of symptoms is rather weak; authors have to itemize how physicians have stated the lack of symptoms: they have to specified if they refer to anamnestically detected symptoms, since we can’t find in the text any concern to a stress test. Due to the gradual progression of the stenosis and patient’s adaptation, it could be difficult recognize symptoms onset; moreover, patients tend to deny the presence of symptoms or they could attribute them to the ageing process or other comorbidities. For these reasons in current guidelines exercise testing is recommended in physically active patients for unmasking symptoms and for risk stratification of asymptomatic patients with severe aortic stenosis; furthermore exercise stress echocardiography may provide prognostic information in asymptomatic severe aortic stenosis by assessing the increase in mean pressure gradient and change in LV function during exercise test. 

We agree, and to accommodate the reviewers comment we have clarified under Methods section: 

“The patients who were categorized as asymptomatic were identified based on information from the electronic patient journal. In most cases, the evaluation of symptoms was based on patient history. Only 13 patients (11 %) had a cardiopulmonary exercise test.”

Under limitations we have written that the evaluation of symptoms was encumbered with uncertainty:

“The reliability of symptom assessment is always associated with uncertainty” and “Furthermore, only a minority of the patients had an exercise test.”¨

- Baseline characteristics of study population should include the reason of admission to hospital 

All patients were referred to our tertiary centre either from their local hospital or an outpatient clinic for elective evaluation for surgery for aortic stenosis. We should have stated this more clearly in the text and have added the following sentence under the methods section where we state that the patients were electively admitted:

“By reviewing the patients’ medical records we identified every patient who had been electively admitted to our tertiary center for evaluation for AVR due to severe aortic stenosis.”

- Authors should explain how they perform follow-up evaluation; in this group of patients with severe aortic stenosis a close follow-up is mandatory; a scheduled visit could unmask symptoms onset and detect some clinical or bio-humoral changes and possible change therapeutic strategy. According to this porpoise one-year survival between two groups (age and gender matched) is similar: 89% in asymptomatic patients compared with 92% in patients referred to AVR. Moreover, a subsequent echocardiogram could reveal an increase of the mean gradient, an incipient left ventricular dysfunction and make mandatory a surgical indication. 

We agree with the reviewer that a thorough follow-up is important in these patients. When declined from surgery, the patients are referred back to their local hospital for further follow-up. We have included the following information in the revised manuscript: 

“Patients who were refused from surgery at Oslo University hospital were referred back to their local hospitals for further follow-up including routine echocardiography in accordance with prevailing guidelines.”

- Likewise authors recordered causes of death in the asymptomatic aortic stenosis group, they should list also causes of death in patient submitted to AVR. 

We thank he reviewer for this input and we have included the following information in the revised manuscript:

“Thirty patients who had AVR died during follow up. Ten deaths (33 %) were cardiovascular, Eight died of malignancy, and six of infections. Two patients died of aortic dissection/aneurism, one of dementia, one of stroke, and two of bowel disease.” 

- In competing risk regression analysis authors should add also troponin between variables at issue. 

For the competing risk regression analysis for both mortality and MACE we used a predetermined model as recommended in the literature (2). We selected parameters that, based on previous research, have been shown to be associated with outcome. These multivariable models were conceived when the study was designed. We discussed whether Troponin T should be included, but decided to avoid including two biomarkers that were likely to be correlated (NT-proBNP and Troponin T). In addition, previous history of coronary artery disease would also likely be correlated with Troponin T. Furthermore, due to the limited number of patients in our study sample, there would be a risk of overfitting the model if we included too many variables. Due to the notions mentioned above, we have decided to keep to our predetermined model and not include Troponin T to the competing risk regression analyses in our revised manuscript.

2. Babyak MA. What you see may not be what you get: a brief, nontechnical introduction to overfitting in regression-type models. Psychosomatic medicine. 2004;66(3):411-21.

Minor points 

- Title does not fully reflect main topics covered in the study; in particular patient characteristics are not been adequately investigated. 

We agree with this comment and have changed the title accordingly to 

“Survival in asymptomatic severe aortic stenosis” 

- Some sections have not a proper layout. 

We have tried to comply with the guidelines by PLOS One and we are sorry to say that we do not understand what the reviewer means by this. We have also made alterations to the methods section according to the reviewers’ comments. We would greatly appreciate it if you would let us know if any part is still insufficient so we can edit accordingly. 

- Conclusions need to be reformulated. 

We have changed the conclusions to only containing the results from our study so we don’t over interpret our results:

“The optimal management strategy for asymptomatic patients with severe aortic stenosis remains unclear. In our retrospective study, patients who were advised against surgery at our tertiary care hospital due to a perceived lack of symptoms had significantly higher mortality than patients referred for AVR.”

---

## [Decision Letter · Decision Letter 1]

22 Mar 2021

Outcomes in asymptomatic, severe aortic stenosis

PONE-D-20-37902R1

Dear Dr. Kvaslerud,

We’re pleased to inform you that your manuscript has been judged scientifically suitable for publication and will be formally accepted for publication once it meets all outstanding technical requirements.

Kind regards,

Claudio Passino, MD

Academic Editor

PLOS ONE

Additional Editor Comments (optional):

Reviewers' comments:

Reviewer's Responses to Questions

**Comments to the Author**

1. If the authors have adequately addressed your comments raised in a previous round of review and you feel that this manuscript is now acceptable for publication, you may indicate that here to bypass the “Comments to the Author” section, enter your conflict of interest statement in the “Confidential to Editor” section, and submit your "Accept" recommendation.

Reviewer #1: All comments have been addressed

Reviewer #2: All comments have been addressed

2. Is the manuscript technically sound, and do the data support the conclusions?

Reviewer #1: Yes

Reviewer #2: Yes

3. Has the statistical analysis been performed appropriately and rigorously? 

Reviewer #1: Yes

Reviewer #2: Yes

4. Have the authors made all data underlying the findings in their manuscript fully available?

Reviewer #1: Yes

Reviewer #2: Yes

5. Is the manuscript presented in an intelligible fashion and written in standard English?

Reviewer #1: Yes

Reviewer #2: Yes

6. Review Comments to the Author

Reviewer #1: The authors have adequately addressed all my concerns. This was a very nice study and is important that the manuscript be disseminated to the scientific community.

Reviewer #2: Authors have sufficiently satisfied comments reported;except for evaluation of symptoms, only 11% of patients have a functional evaluation (cardiopulmonary stress test in 13 patients), which has been correctly reported in the new version. Symptoms recognition due to the gradual progression of aortic stenosis and patient’s adaptation could be hard to evaluate; moreover, patients tend to deny the presence of symptoms or they could attribute them to the aging process or other comorbidities. For these reasons in current guidelines exercise testing is recommended in physically active patients for unmasking symptoms and for risk stratification of asymptomatic patients with severe aortic stenosis.

Moreover troponin Ths evaluation in regression analysis would have provided interesting data, not only in the context of coronary heart disease, but especially in cardiomyopathies ambit, such as cardiac amyloidosis, often associated to aortic stenosis.

7. PLOS authors have the option to publish the peer review history of their article (what does this mean?). If published, this will include your full peer review and any attached files.

Reviewer #1: **Yes: **Crochan O'Sullivan

Reviewer #2: No

---

## [Editor Report · Acceptance letter]

26 Mar 2021

PONE-D-20-37902R1 

Outcomes in asymptomatic, severe aortic stenosis 

Dear Dr. Kvaslerud:

I'm pleased to inform you that your manuscript has been deemed suitable for publication in PLOS ONE. Congratulations! Your manuscript is now with our production department. 

Kind regards, 

on behalf of

Prof. Claudio Passino 

Academic Editor

PLOS ONE